# A vertical transport window of water vapor in the troposphere over the Tibetan Plateau with implication for global climate change

**Xiangde Xu[1,2], Chan Sun[1,2], Deliang Chen[3], Tianliang Zhao[1*], Jianjun Xu[4],**

**Shengjun Zhang[2], Juan Li[1], Bin Chen[2], Yang Zhao[2], Hongxiong Xu[2], Lili Dong[2],**

**Xiaoyun Sun[1], Yan Zhu[1]**

[1] Nanjing University of Information Science and Technology, Nanjing 210044, China;

[2] State Key Laboratory of Severe Weather, Chinese Academy of Meteorological Sciences, Beijing, 100081, China;

[3] Regional Climate Group, Department of Earth Sciences, University of Gothenburg, Sweden;

[4] South China Sea Institute of Marine Meteorology, Guangdong Ocean University, Zhanjiang 524088, China

* Corresponding author. **Email:**tlzhao@nuist.edu.cn

**Abstract**

By using the multi-source data of meteorology over recent decades, this study discovered a summertime "hollow wet pool" in the troposphere with a center of high water vapor over Asian water tower (AWT) on the Tibetan Plateau (TP), which is featured by a vertical transport "window" in the troposphere. The water vapor transport in the upper troposphere extends from  the vertical transport window over the TP with the significant connections among the Arctic, Antarctic and TP regions, highlighting the effect of TP's vertical transport window of  water vapor in the troposphere on global change of water vapor. The vertical transport window is built by the AWT's thermal forcing  in association with the dynamic effect of the TP's "hollow heat island". Our study improves the understanding on the vapor transport  over the TP with an important implication to global climate change.


## 1. Introduction

**1. Introduction**
The Tibetan Plateau (TP) is the largest high terrain in the world, known as "the roof
of the world" with an averaged altitude over 4,000 meters. The rivers, such as the
Yangtze River, Yellow River, Lancang River and Ganges River, are all originated from
the TP, which is regarded as the "Asian Water Tower" (AWT) (Xu et al., 2008). The
three-river-source (Yangtze, Yellow, and Lancang Rivers) region (TRSR) in the eastern
TP is the core area of the AWT over the plateau (Xu et al., 2014). The observed "CISK-
like mechanism" is an important mechanism sustaining the atmospheric "water tower"
over the AWT (Xu et al., 2014). Connecting with the cloud and precipitation in the AWT,
the plausible hydrological cycles could be realized with the transport of water vapor from
tropical oceans up to the TP (Xu et al., 2014).

Water vapor plays an important role in global environment and climate changes
(Tian et al.,2009; Solomon et al.,2010). The ratio of strong convective clouds to total
clouds over the Tibetan Plateau (TP) is about 5 times to the global ratio, and the frequent
occurrences of strong convective clouds could be largely attributed to the TP's large
topography (Luo et al.,2011; Su et al.,2006). The water vapor in the upper troposphere is
mainly originated from the tropical lower troposphere through vertical transport and
evaporation of convectively transported or in situ produced cloud ices (Tian et al.,2004;
James, et al.,2008). Water vapor was first lifted by convection over the Bay of Bengal
and the South China Sea and then transported upwards the tropical tropopause layer via
the monsoon anticyclonic circulations towards Northwest India (Yanai, et al., 1973; Chen,
et al., 2012). TP is a moisture sink in summer, having a net moisture convergence of 4
mm each day, where the convergences were enhanced from 1979 to 2018 (Feng and
Zhou, 2012; Xu, et al., 2020).   In general, Asian monsoon circulation provides an
effective pathway for regional water vapor transport to the TP (Wang, et al.,2017). An
important role of the anticyclone over the TP is verified in the exchange of water vapor
between the troposphere and stratosphere (Garny, et al., 2016; Fu, et al., 2006) . Many
studies have been focused on the transport of water vapor into upper troposphere and
lower stratosphere from the tropical oceans to the high-altitude TP (Chen, et al., 2012;
Wang, et al.,2017; Xie, et al.,2018; Randel, et al.,2013). However, inadequate attention
has been paid to the vertical transport of water vapor in the troposphere over the TP,
especially in respect of the underlying meachnism and the consequences on global
climate.

The following questions are of great concern in the TP' vertical transport of water

vapor study with the implication for global change, for example, what is the formation
mechanism on the vertical transport window of water vapor in the troposphere on the TP?
How is the vertical transport of water vapor in the troposphere constructed with the
special column of apparent heat source in the AWT over the TP ? How is the global
effect of the vertical transport window of water vapor in the troposphere on the TP?
From the perspective of global atmospheric energy and water vapor exchanges, this study
characterizes a window of water vapor vertical transport within the troposphere over the
TP and the implication for global change.

## 2. Data and Methods

The daily meteorological data of cloud amount are provided by the meteorological observatories in the TP in the period of 1979 to 2018.The AIRS remote sensing products of water vapor from 2003 to 2018 and the ECMWF-interim data of meteorology from 1979 to 2018 are used in this study.

In this study, the inverse algorithm is used to calculate the apparent heat source $Q_1$, and the formula is as follows ( Su et al.,2006) :

$$Q_1 = C_p[\frac{\partial T}{\partial t} + \vec{V} \cdot \nabla T + (\frac{p}{p_0})^k \overline{\omega \frac{\partial \theta}{\partial p}}] \tag{1}$$

where $T$ is the air temperature; $\omega$ is the vertical velocity at the $p$ coordinate, $P_0 = 1000\,\mathrm{hPa}$; $k = R/C_p$ ; $V$ is the horizontal wind vector; $\theta$ is the potential temperature.

Vertical integration of $Q_1$ is expressed as:

$$\langle Q_1 \rangle = \frac{1}{g} \int_{p_t}^{p_s} Q_1 \, dp \tag{2}$$

where $p_s$ is the surface air pressure, and $p_t$ is the top air pressure, here taken as 300hPa.

In order to analyze the relationship between water vapor sources and vapor transport channels in the atmospheric water cycle, the correlation vector calculation was used to calculate the temporal and spatial variations of the water vapor transport channels. The expression is:

$$\vec{R}(x, y) = R_u(x, y)i + R_v(x, y)j \qquad\qquad (3)$$


where $\vec{R}(x,y)$ represents the correlation vector in which $R_u\,(x,\,y)$ represents the


correlation coefficients between water vapor and the component of latitudinal water


vapor flux $qu$, and $R_v\,(x,\,y)$ represents correlation coefficients between water vapor and


longitudinal water vapor flux components $qv$.



## 3. Results and discussion


### 3.1 The structures of vertical transport window of water vapor over the TP


With the use of satellite remote sensing products from 2003 to 2016, the global


distribution of the total water vapor from 500hPa to 300hPa in the troposphere was


shown in Figure 1a . The results indicate that there is a high value center of water vapor


in the mid- and upper troposphere over the TP, extending southwards to the Bay of


Bengal, India and Northern Southeast Asia. It is worth noting that the fraction of strong


convective clouds to the total cloud ranges from 4.0 % to 21.0 % in the TP, and during


the summer season the thermal forcing of TP is dominated by the latent heat released by


cloud and precipitation (Fu et al.,2006; Dessler et al.,2006; Gao et al.,2014). The intense


mesoscale convective activity, which is represented with the low cloud fraction based on


the could characteristics observed in the TP, and the "massive chimney effect" of huge


cumulonimbus cloud drive the transport of atmospheric heat and water vapor to the upper


troposphere (Fu et al.,2006; Xie et al.,2018). Based on Chinese Third Tibetan Plateau


Experiment-Observation of Boundary Layer and Troposphere (2014–2017), it is observed


in the TP that the cloud-top height was averaged around 11.5 km (a.s.l.) with its


maximum value exceeding 19 km (a.s.l.) , and the mean cloud-base height at 6.88 km
(a.s.l.) during the observation period, reflecting the TP's deep convection in the
troposphere and its  impact on the upper troposphere.

**3.2 Global effect of the vertical transport window over the TP**
The vertical section of the correlation coefficients  along the south-north direction
between the low cloud cover on  the TP and the global water vapor are presented in
Figure 1b. The obviously upward movement of water vapor over the TP can be seen in
Figures 2a. It could be noticed that there exist the structures similar with the massive
chimney  between the convective cloud and the water vapor on the TP (Figures 1b and
2a). Figures 2b and 2c show significant correlation between convective clouds over the
AWT and the changes of global water vapor from 1979 to 2018. Significant correlations
extend from the TP southward and northward in the upper troposphere. It is remarkable
that the high correlation areas passing the 90% confidence level expand towards the polar
regions of both the southern and the northern hemispheres (Figures 1b, 2b and 2c),
depicting the relation between the convective clouds and the global water vapor in the
upper troposphere across the northern and southern hemispheres for an implication of the
TP to global climate change.
The distributions of high positive correlation coefficients between low cloud
cover over the TP and the global water vapor in the upper troposphere are calculated by
ECMWF-interim reanalysis data (Figure 3a). It can be found that there is a region with
highest values of correlation coefficients in the upper troposphere (500hPa-300hPa),
covering a banded large area from the plateau across the lower latitude tropical zone to
the polar regions, indicating the significant correlations between convective cloud
activities on the TP and the global water vapor in the upper troposphere, especially in the
polar region of the southern hemisphere area (Figure 3a),which could be reflected an
importance of the thermal forcing of TP in global changes of water vapor.
The strong anticyclone in the upper troposphere over the southeastern TP takes a
significant part in the upward transport of water vapor in the troposphere and stratosphere
( Garny, et al., 2016;Fu, et al., 2006).  In order to understand the effect of the vertical
transport window of troposphere over the TP on the global water vapor distribution from
the perspective of the dynamic effect of anticyclone over the plateau driven by the heat
sources, we presented the distributions of correlation coefficients between daily mean $Q_1$
in the TP and global water vapor flux in July from 2014 to 2016 at 300hPa (fig. 3b.)
Driven by the heat source of the TP, the anticyclone is formed in the upper troposphere
over the TP and surrounding regions, which governed the water vapor transport form the
TP not only to the surrounding area, but also extending to the north and south poles along
the long-range transport channels (Fig. 3b), which indicates the vertical transport window
effect of the TP on global water vapor transport, especially over high-latitude regions
such as the Arctic and Antarctic. To further verify the global transport pathways of water
vapor from the TP, we used the methods of composite analysis to characterize global
distribution of water vapor transport fluxes at the 300hpa in the years to anomalously
high and low $Q_1$ over the TP. The TP's anticyclone in the upper troposphere is often
associated with deep convection in the troposphere (Garny, et al., 2016). Fig. 3c
shows that in years with higher $Q_1$, stronger anticyclone formed at the upper troposphere
(Fig. 3b), which maintains the upward transport of water vapor to the upper troposphere,
with strong transport of water vapor transport the arctic and antarctic (Fig. 3c),
confirming the impact of the vertical transport in the troposphere driven by heat released
within AWT in the TP on global water vapor transport especially to the polar regions.
The Indian continent heats up from spring to summer, hence the convection draws
moisture northwards from the Bay of Bengal, Arabian Sea and Indian Ocean, leading to
precipitation in the Himalayas and beyond (Yanai et al., 1973). In Figure 3d, it could be
found that, driven by the strong apparent heat source, the water vapor flows from the low
latitude ocean could build a remarkable channel to the TP. The key entrance to the water
vapor passage is just the intersection of the Himalayas on the southern slope of the TP.
This region constitutes a special canyon pass in the plateau with deep valleys, making a
perfect entrance zone for the oceanic warm-wet water flows (see theterrain distribution
inserted in the lower right corner of Figure 3d).
FLEXPART trajectory model (Stohl, et al., 2005;Reale,et al 2001; James, et al, 2004)
was used to simulate the spatial and temporal changes of water vapor transport to the
TRSR over the TP, driven with the ERA-Interim reanalysis data of meteorology with
horizontal resolution of $0.75^{\circ} \times 0.75^{\circ}$ in July 2009. In the FLEXPART particle diffusion
model, the 80000 particles was released at the TRSR ($90^{\circ}$ -$102^{\circ}$ E and $30^{\circ}$ -$35^{\circ}$ N). In
Figure 3f, it can be found that the water vapor in the TRSR was traced to water vapor
source on the tropical Indian Ocean. The water vapor from the central Indian Ocean in
the southern hemisphere can be transported along the Somali jet flow through the
Arabian Sea to the TP. The water vapor from the South China Sea and the Bay of Bengal
was transported to the TP converging over the TRSR (Figure 3f), characterizing the water
vapor transport channel from the southern hemispheric and low latitude oceans to the TP.
According to the correlation analysis of water vapor transport, the water vapor
source of the AWT can also be traced back to the ocean surface water vapor source
region with water vapor positive correlation extreme value region in the Chagos
Archipelago of the Central Indian Ocean near 10°S south of the equator (Figure 3e),
revealing that the TP is the confluence area of the  hemispherical water vapor from the
southern Indian Ocean.

**3.3 The transport window of water vapor driven by the AWT**
Through the correlation analysis of the column apparent heat $Q_1$ over the TP as
well as the three-dimensional structure of vorticity and divergence, it can be found that
the apparent heat source $Q_1$ in the TP is an important forcing factor (Figure 4). The
results show that the air heat island in the AWT is located  at 300-500 hPa in the upper
tropopshere, which is regarded as the high apparent heat $Q_1$ area significantly related to
the convective clouds and the strong ascending movement (Figures 4a and 4d). Figures
4b,4c. 4e and ff present the correlations of the column apparent heat $Q_1$ in AWT with the
divergence and vorticity fields over the TP, which can describe the effective "suction
effect"  with divergence (negative vorticity) at  upper levels and convergence (positive
vorticity) at lower levels in the troposphere. The $Q_1$ is significantly released in the
convective clouds and the strong ascending movement, and there exists a strong
anticyclonic circulation in the upper troposphere over the region of the AWT in the
southeast of the plateau (Figure 3b). In addition, the lower troposphere is the center of
strong convergence and strong vorticity.  Figure 3g shows the difference of vapor
transport flux and specific humidity at 500hPa in summer between anomalously high and

low $Q_1$. When the $Q_1$ in TRSR is anomalously high, large water vapor from the tropical oceans is transported across the Bay of Bengal and the Indian peninsula, and entered the TP from the southern edge, revealing the TP's thermal effect could make a strong vapor transport channel connecting the water vapor source in the low latitude tropical oceans.

All these results reveal the effective "pumping effect" of the vertical configuration with low-level cyclonic circulation and high-level divergence with anticyclone circulation over the TP. The strong confluence effect building the vertical transport window of water vapor could be driven by the elevated heating on the TP in the troposphere with the water vapor flow, making a strong vapor transport connecting the water vapor source in the low latitude oceans with the high water vapor center over the core area of AWT over the TP. The water vapor transport connect from the vertical transport window over the TP and the Arctic, Antarctic regions in the upper troposphere, highlighting the effect of TP "hollow wet pool" on global climate change.

**4. Conclusion**

By using the multi-source data of meteorology over recent decades, this study discovered a summertime "hollow wet pool" in the troposphere with a center of high water vapor over AWT on the highly elevated TP, which is featured by a vertical transport window with the transport flux of water vapor in the troposphere. Driven by the strong TP's heat source, water vapor flows connect the AWT over the TP with the low-latitude oceans. Significant correlations exist between convective activities on the TP and global water vapor in the upper troposphere,. The water vapor transport from the TP's

vertcial window in the upper troposphere extends from the TP globally towards the
northern and southern hemispheres with the significant connections among the three
poles of Arctic, Antarctic and TP regions, highlighting the effect of TP's vertical
transport window of water vapor on global climate change. The vertical transport window
is built by the AWT's thermal forcing in association with the dynamic effect of the TP's
"hollow heat island" as well as the effective "pumping effect" on vertical transport with
low-level convergences with cyclonic circulation and upper-level divergences with
anticyclone circulation in the troposhere over the TP.

Basd on this observational study, a conceptual model of the comprehensive relation

of the TP region with the global energy and water cycles is put forward for the vetical
transport window of vapor in the troposphere driven by the thermal forcing in the core
region of the AWT over the TP (Figure 5), where the water vapor source is traced back to
tropical ocenas and the Southern Hemisphere. The thermal effect of the TP could sustain
the vertical upward transport of the energy and water vapor. The water cycle in the AWT
clearly displays the linkages of the vertical transport window of water vapor in the
troposphere over the TP  with the vapor source in the tropical oceans and the southern
Indian Ocean in the lower troposphere and with the Arctic and Antarctic regions in the
upper troposphere ((Figure 5). Our study depicts a comprehensive understanding on the
vertical water vapor transport in the atmosphere over the TP with an important
implication to global climate change.

***Data availability.*** ERA-Interim of ECMWF (https://apps.ecmwf.int/datasets/data/interim-
full-moda/levtype=pl/) reanalysis daily and  monthly data are part of the European Center
for Medium-range Weather Forecasts. AIRS Science Team/Joao Teixeira (2013),
AIRS/Aqua L3 Daily Standard Physical Retrieval (AIRS-only) 1 degree x 1 degree V006,
Greenbelt, MD, USA, Goddard Earth Sciences Data and Information Services Center
(GES DISC), Accessed: [Jan. 2019], 10.5067/Aqua/AIRS/DATA303. The low cloud data
used in this study are derived from the Data Sets of Surface Meteorological Elements in
China released by the National Meteorology Information Center, China Meteorological
Administration,which can be found at
https://zenodo.org/record/5121157#.YPkRHqjitPY (Sun, 2021).

**Author Contributions.** XDX, CS and TLZ conducted the study design. DLC, JJX and
SJZ analysed the observational data. JL, BC, YZ, HXX, LLD, XYS, and YZ assisted
with data processing. XDX, CS and TLZ wrote and revised the manuscript. XDX, CS,
TLZ, and JJX were involved in the scientific interpretation and discussion. All authors
provided commentary on the paper.

**Competing interests.** The authors declare that they have no conflict of interest.

**Acknowledgments.** The authors acknowledge the support from the  The Second Tibetan
Plateau Scientific Expedition and Research (STEP) program and the Scientific and
Technological Development Funds from Chinese Academy of Meteorological Sciences.
We would like to thank the editor and the two anonymous reviewers, for their
suggestions on restructuring the initial draft.
**Financial support.** This study was supported by The Second Tibetan Plateau Scientific
Expedition and Research (STEP) program (2019QZKK0105) and the Scientific and
Technological Development Funds from Chinese Academy of Meteorological Sciences
(2021KJ022 and 2021KJ013).

**Competing interests.** The authors declare that they have no conflict of interest.

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

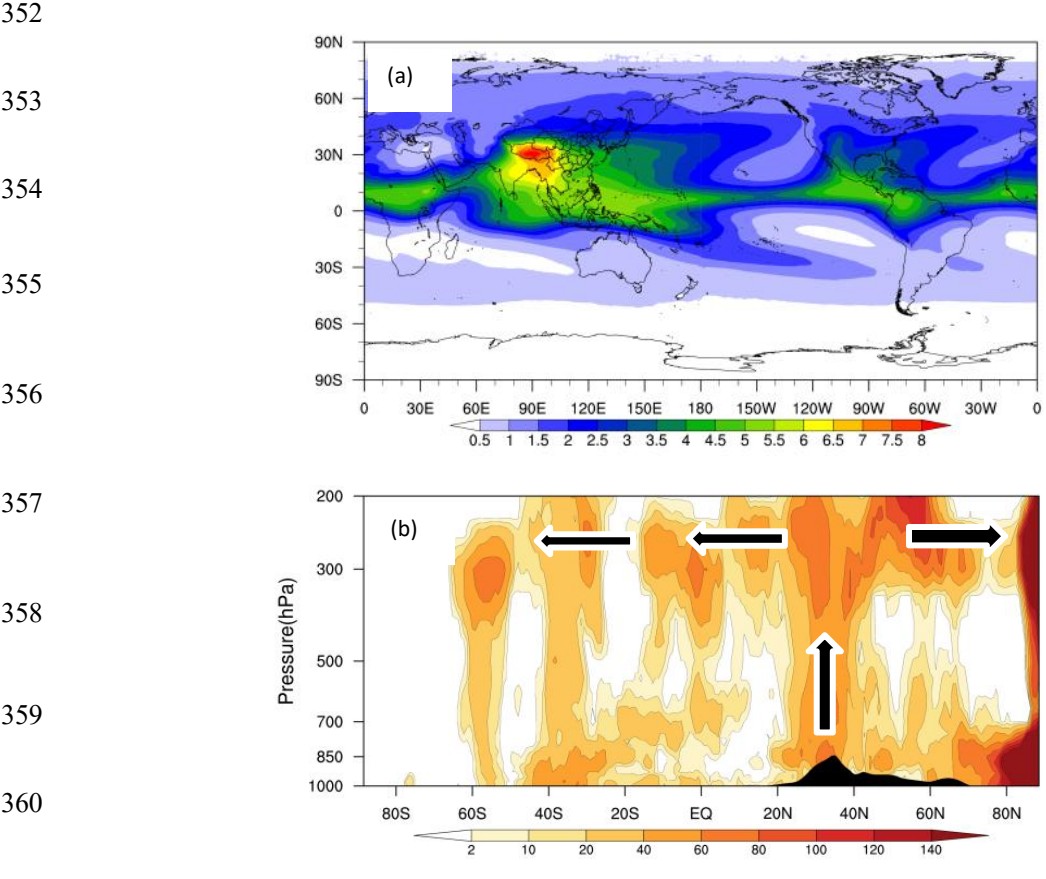

**Figure 1.** (a) The global distribution of the total water vapor from 300 hPa to 500 hPa based on the summertime AIRS data from 2003 to 2018, (b) the vertical section of the frequency (shaded) of the correlation coefficients passing the level of 90% confidence between summertime TP's low cloud cover and the water vapor at different vertical levels along the meridional direction averaged over 60ºE - 180ºE for 1979-2016 with the black arrows indicating the connections of TP's low clouds to global water vapor in the upper troposphere with high frequencies.

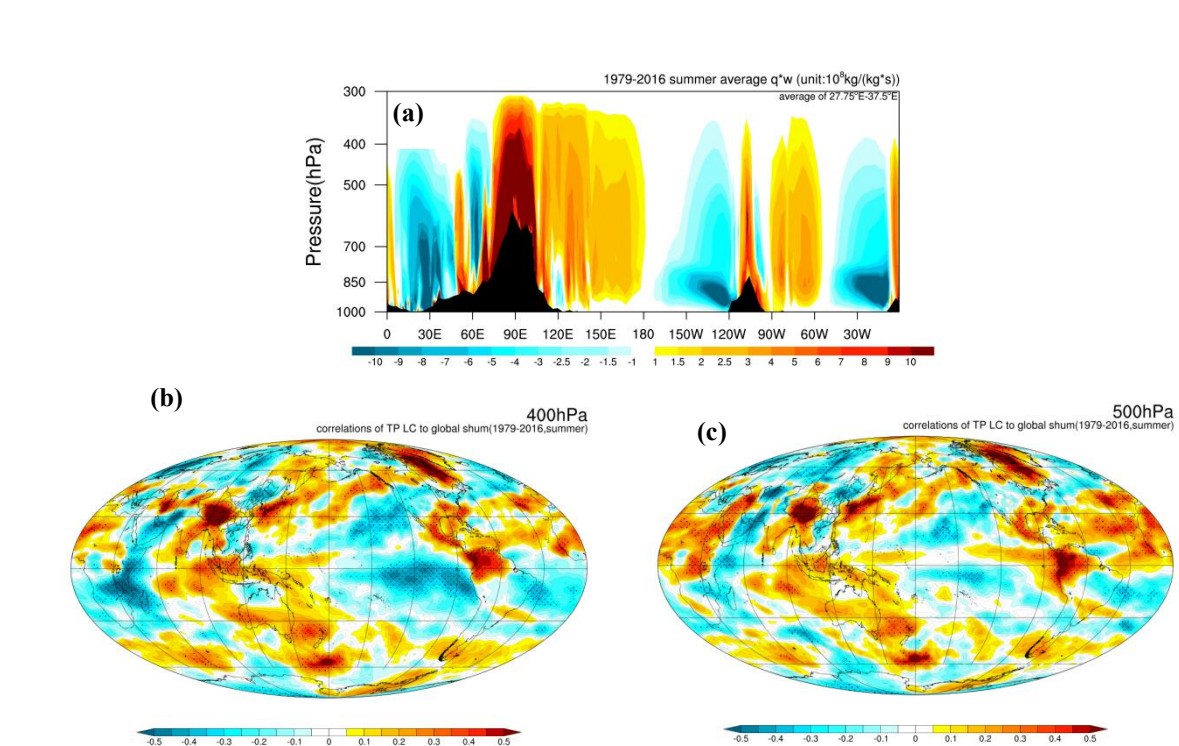

**Figure 2.** (a) The vertical section of vertical vapor transport flux averaged over 27.5-32.0°N in summers of 1979-2016; the spatial distributions of correlation coefficients of low cloud cover over the TP with  the global specific humidity of the ECMWF-interim data in Summer (June, July and August) from 1979 to 2018 at (b) 400hPa and (c) 500hPa.

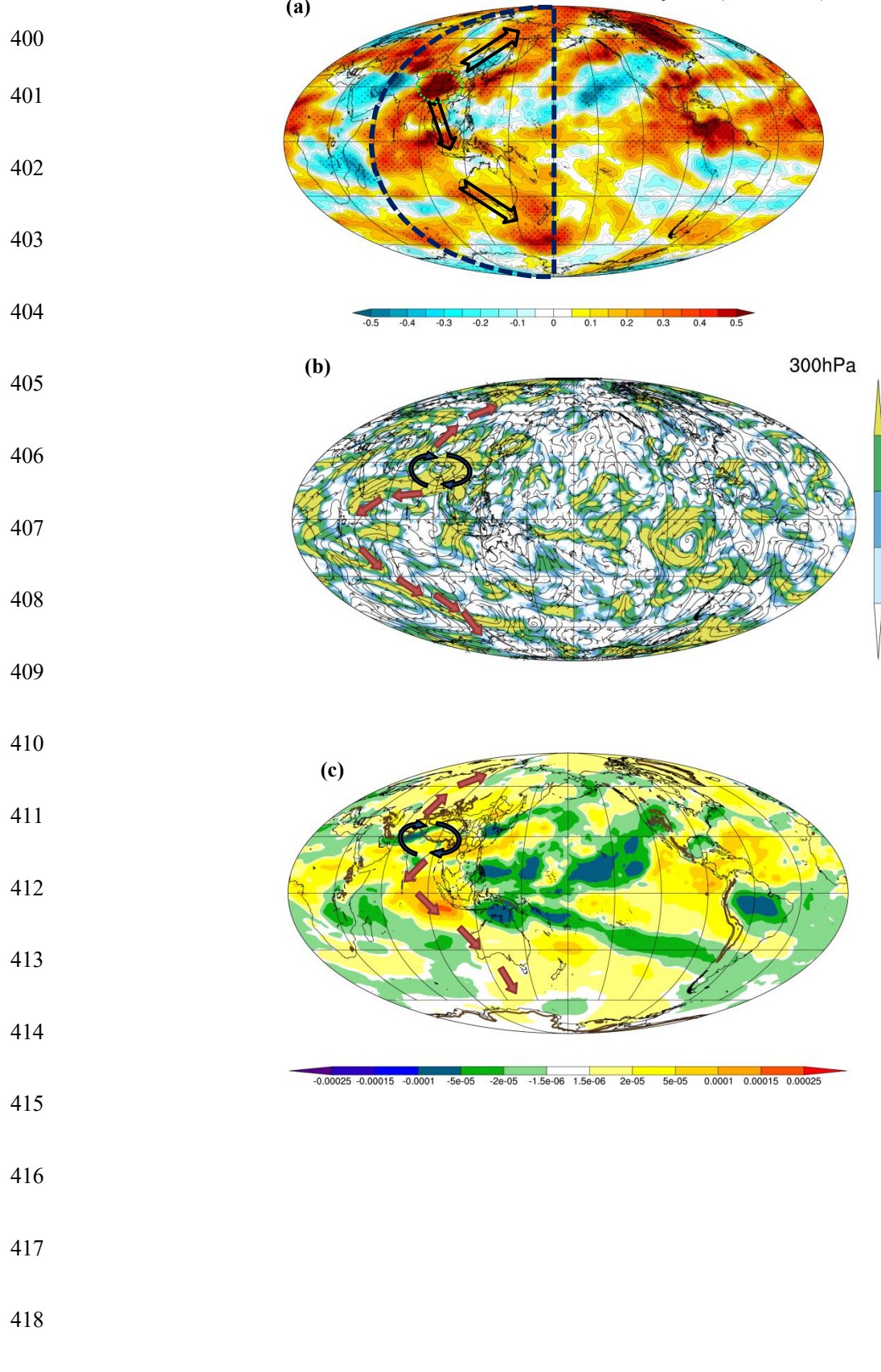



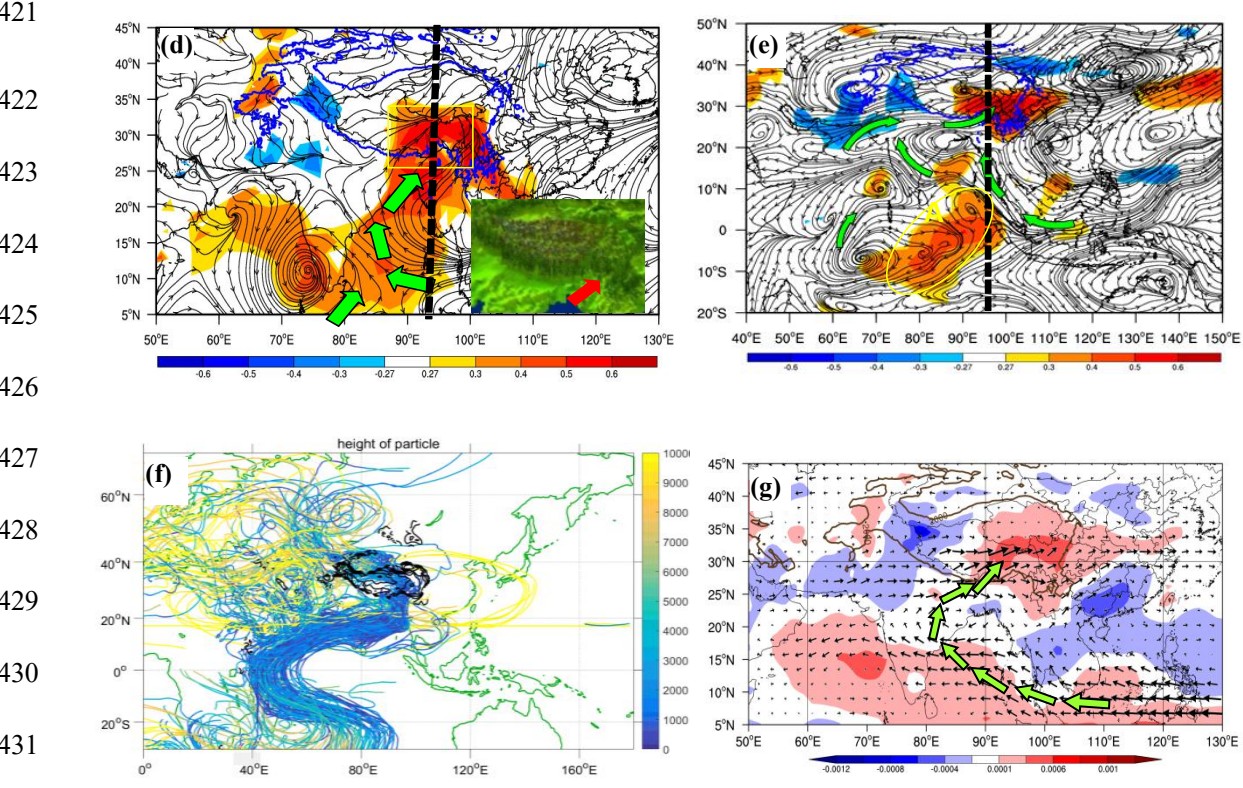












**Figure 3.** (a) The spatial distributions of correlation coefficients of low cloud cover over the TP with the global specific humidity of the ECMWF-interim data at 300 hPa in summers of 1979-2016 with the pathways of convective air to the troposphere; (b) correlation vectors of the column $Q_1$ integrated vertically over the TP region (80-102°E; 30-37.5°N) with the 300hPa vapor transport flux in July of 2014-2016,The shaded area indicates the correlation coefficient passing the the the 90% confidence level; (c) the difference of specific humidity (shading, unit:kg/kg) at 300 hPa in summer in 1998 and 2007 with anomalously high $Q_1$ and in 1997 and 2003 with anomalously low $Q_1$ in the AWT, The black and orange arrows indicate respectively the anticyclonic circulations in the TP and water vapor ttransport pathways from the TP to the Arctic and Antarctic regions.; the correlation field between the total apparent heat source Q1 over the TP region (80-102°E; 30-37.5°N) with the water vapor (shaded) and water vapor flux (stream lines)

in  the surface layer (d) and middle layer (500hpa) (e) in summer over 1979-2015, respectively, (f)
the backward trajectories of water vapor transport simulated with the model FLEXPART in July,
2009. (g) the difference of vapor transport flux at 500 hPa (vectors, unit:gs$^{-1}$hPa$^{-1}$cm$^{-1}$) and
specific humidity (color contours, unit:kg/kg) between summers with anomalously high $Q_1$ in
1998, 2005, 2007, 2008 and 2009 and with anomalously low $Q_1$ in 1994, 1997, 2001, 2002 and
2003 over the TP

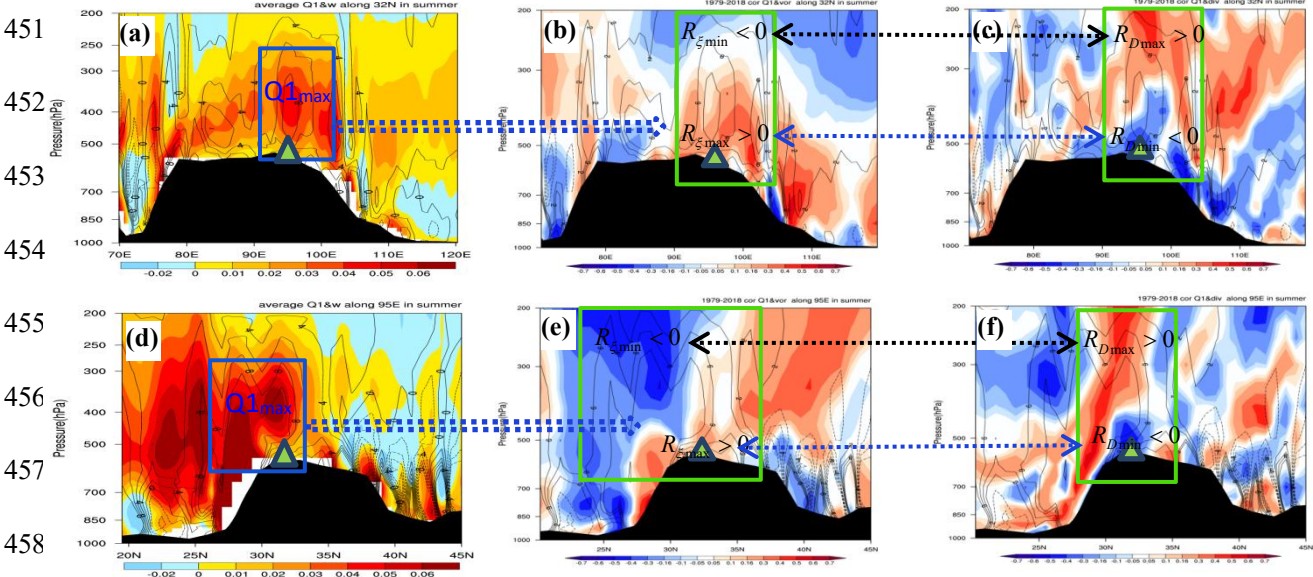


**Figure 4.**The vertical sections of (a,d) vertical motion (contours, in unit: $10^{-2}$Pa·s$^{-1}$) and $Q_1$
(color contours, in unit:$10^{-3}$w kg$^{-1}$); (b,e) vertical motion (contours, in unit: $10^{-2}$Pa·s$^{-1}$) and
correlation coefficients (color contours) between $Q_1$ and the vorticity as well as (c,f) vertical
motion (contours, in unit: $10^{-2}$Pa·s$^{-1}$) and the correlation coefficients between $Q_1$ and the
divergence (contours) in the TP, with Figs. a, b and c  along 32 °N, and Figs. d, e and f along 95
°E. The green triangles indicate  the AWT core region.




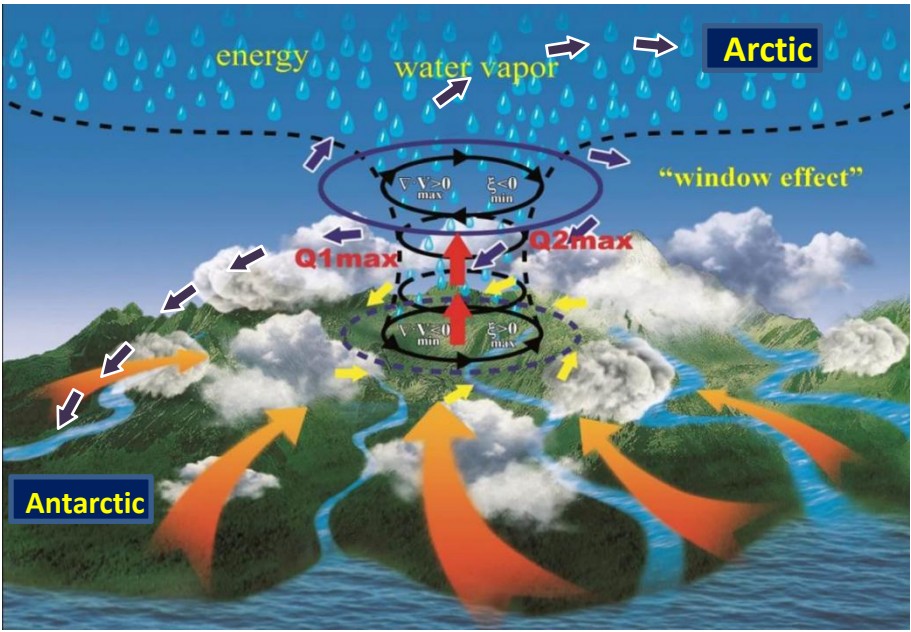


**Figure 5.** A diagram of vertical water vapor transport in the troposphere driven by the thermal forcing of AWT over the TP, where the vertical transport window of water vapor in the troposphere connects globally the water vapor transport from the tropical oceans and the southern Indian Ocean in the lower troposphere with transport to the Arctic and Antarctic regions in the upper troposhere.

476

477

478