# Peer review of "A vertical transport window of water vapor in the troposphere"

_Atmospheric Chemistry and Physics, 2021_

## Author Comment (AC1)

**Reply to Referee 1**

**We are grateful to the referee for the encouraging comments and careful reviews which helped to improve the quality of our paper. In the followings we quoted each review question in the square brackets and presented our response after each paragraph.**

*[Review Comment: The manuscript revealed the forcing mechanism forming the vertical transport window of water vapor in the troposphere on the TP. It characterizes a window of water vapor vertical transport within the troposphere over the TP and the implication for global change. This work is very meaningful and the paper has been well-written. I therefore recommend this paper resubmitted after minor revisions. ]*

**Reply:** Thank you for the encouraging comments.

*[1. Figure 1b is about the frequency of the correlation coefficients passing the level of 90% confidence between summertime TP's low cloud cover and the water vapor at different vertical levels. How do authors get the frequency? Please give the specific introduction of it.]*

**Reply:** Frequency here refers to the number of points passing the significance test on the same latitude between 60ºE - 180ºE .

*[2. Figures 2b and 2c are the spatial distributions of lag correlation coefficients. From the caption and related analysis, I didn't get the meaning of lag correlation coefficients. In the result section, there is no any analysis and discussion about the Figures 2b and 2c. Please add more illustration and discussion.]*

**Reply:** Sorry about the description of *lag correlation coefficients*. Figure 2b and 2c

show the spatial distributions of correlation coefficients of low cloud cover over the TP and the global specific humidity in the same month in summer (June, July and August separately) from 1979 to 2018 at (b) 400 hPa and (c) 500 hPa. We have rewritten the description and added the illustration in the manuscript as follow:

"The vertical section of the correlation coefficients along the south-north direction between the low cloud cover on the TP and the global water vapor are presented in Figure 1b. The obviously upward movement of water vapor over the AWT can be seen in Figure 2a. It could be noticed that there exist the structures similar with the massive chimney between the convective cloud and the water vapor on the TP. Figure 2b and 2c show significant correlation between convective clouds over the AWT and water vapor over the region. Such a significant correlation began to extend southward and northward at 400~500hPa. It is remarkable that the high correlation areas exceeding the 90% confidence level expand towards the polar regions of both the southern and the northern hemispheres (Figure 1b), and the relation between the convective clouds and the global water vapor in the upper troposphere across the northern and southern hemispheres could be depicted. "

*[3. As seen from Figure 4, it contains lots of information, but the related analysis is too simply. Please add more analysis and discussions.]*

**Reply:** We added two subgraphs in Figure 4. The new figure and description was adjusted as follow:

[Figure]

**Figure 4.** The vertical sections of vertical motion (contours, in unit: $10^{-2}$Pa·s$^{-1}$) and average Q1(shaded, in unit:$10^{-3}$w/kg)(a,d) ;vertical motion (contours, in unit: $10^{-2}$Pa·s$^{-1}$) and correlation coefficients (shaded) between Q1 and the vorticity (b,e) as well as the correlation coefficients between Q1 and the divergence (c,f) separately in the core region of the AWT, in which, a, b, c is along 32 °N, and d, e, f is along 95 °E. The green triangle is the AWT.

We have added the analysis in the manuscript as follow:

" Through the correlation analysis of the whole layer of apparent heat source Q1 over the plateau region, the three-dimensional structure of vorticity and divergence, it can be found that the apparent heat source Q1 in the TP is an important forcing factor (Figure 4). The AWT is located in the mid-high level at 300-500 hPa, which is regarded as the extreme apparent heat source Q1 area, and it is significantly related to the convective cloud and its strong ascending movement(Figure 4a,d). Figure 4b,c,e and f show the correlation between the total apparent heat source Q1 in AWT and divergence/vorticity fields, which can describe the effective "suction effect" that displays the configuration with divergence (negative vorticity) at the upper levels and

convergence (positive vorticity) at lower levels. The Q1 is significantly related to the convective cloud and its strong ascending movement, and there exists a strong high-level anticyclone in the region of the AWT in the southeast of the plateau (Figure 3d). In addition, the lower troposphere is the center of strong convergence and strong vorticity. All these results reveal the effective "pumping effect" of the vertical configuration with low-level cyclonic circulation and high-level divergence with anticyclone circulation in the TP (Figure 4b,c,e,f). The strong confluence effect could be driven by the elevated heating on the TP in the middle troposphere with the water vapor flow, making a strong warm wet vapor transport channel connecting the water vapor source in the low latitude tropical ocean with the water vapor center over the core area of AWT. "

*[4. ~9. L70, tropophere-> troposphere*

*L88, 100hpa--> 100 hPa*

*L307, 60oE - 180oE--> 60oE - 180oE*

*L142, the Asian water tower (AWT) --> AWT*

*L149, Figure 3c should be Figure 3b*

*L150, Figure 3d should be Figure 3c.]*

**Reply:** Following this comment, we have adjusted it as required.

*[10. P17, what does the shading mean in Figures 3b,c and d? What's the difference between Figure 3d and Figures 3b and c?   The correlation in Figures 3b and c are based on the period of*

*1979-2016, aren't they? And why the correlation based on the period of 2014-2016 are given in*

*particular?]*

**Reply:** The shaded parts in Figure 3b and 3c indicate correlation coefficients of

TP-column Q1 integrated to water vapor. The correlation coefficient in Figure 3d

exceeds the significant test at the 90% and more confidence level.

The order of subgraphs in Figure 3 has been changed. The descriptions are adjusted as

follow:

[Figure]

**Figure 3.** (a) The spatial distributions of correlation coefficients of low cloud cover

over the TP with the global specific humidity of the ECMWF-interim data at 300 hPa

in summers of 1979-2016 with the pathways of convective air to the troposphere, (b)

correlation vectors of TP-column Q1 integrated over the TP region (80-102°E; 30-37.5°N) to 300hPa vapor transport flux in July of 2014-2016,The shaded area indicates the correlation coefficient exceeds the significant test at the 90% confidence level; the correlation field between the total apparent heat source Q1 over the TP region (80-102°E; 30-37.5°N) with the water vapor (shaded) and water vapor flux (stream lines) in the surface layer (c) and middle layer (500hpa) (d) in summer over 1979-2015, respectively.

The correlation mentioned in Figures. 3c and 3d is based on monthly mean Q1 and water vapor flux during 1979-2015 in summer. While Figure 3b shows correlation between daily mean Q1 in TP and water vapor flux in July from 2014 to 2016 at 300hPa so as to discuss the driving effect of Q1 on water vapor transport at the synoptic system and process scale in the Plateau region.

We have supplemented the illustration in the manuscript as follow:

"Figure 3b shows the correlation between daily mean Q1 in the TP and water vapor flux in July from 2014 to 2016 at 300hPa so as to discuss the driving effect of Q1 on water vapor transport at the synoptic system and process scale in the Plateau region. From the perspective of daily weather process in July of 2014-2016, the possible mechanism of the global effect of 300hpa anticyclone on water vapor transport is revealed. There exists also a strong high-level anticyclone in the region of the AWT in the southeast of the plateau, which takes a significant part in the exchange of water vapor between the troposphere and stratosphere ( Garny, et al., 2016;Fu, et al., 2006) ."

*[11. P18, the caption makes reader confused. All the contours in these four figures represent the vertical motion, not just in (a). Please rewrite the description of these four subgraphs.]*

Following this comment, we have adjusted it as required.

**"Figure 4.** The vertical sections of vertical motion (contours, in unit: 10-2Pa·s-1) and average Q1(shaded, in unit:10-3w/kg)(a,d) ;vertical motion (contours, in unit: 10-2Pa·s-1) and correlation coefficients (shaded) between Q1 and the vorticity (b,e) as well as the correlation coefficients between Q1 and the divergence (c,f) separately in the core region of the AWT, in which, a, b, c is along 32 °N, and d, e, f is along 95 °E. The green triangle is the AWT."

---

## Author Response (AR1)

**Reply to Referee 1**

**We are grateful to the referee for the encouraging comments and careful reviews which helped to improve the quality of our paper. In the followings we quoted each review question in the square brackets and presented our response after each paragraph.**

*[Review Comment: The manuscript revealed the forcing mechanism forming the vertical transport window of water vapor in the troposphere on the TP. It characterizes a window of water vapor vertical transport within the troposphere over the TP and the implication for global change. This work is very meaningful and the paper has been well-written. I therefore recommend this paper resubmitted after minor revisions. ]*

**Reply:** Thank you for the encouraging comments.

*[1. Figure 1b is about the frequency of the correlation coefficients passing the level of 90% confidence between summertime TP's low cloud cover and the water vapor at different vertical levels. How do authors get the frequency? Please give the specific introduction of it.]*

**Reply:** The correlation coefficient between summertime TP's low cloud cover and the global water vapor at each grid point is calculated at first. Frequency here refers to the number of points, the correlation coefficient of which has passed the significance test (the level of 90% confidence), on the same level and latitude between 60ºE - 180ºE.

*[2. Figures 2b and 2c are the spatial distributions of lag correlation coefficients. From the caption and related analysis, I didn't get the meaning of lag correlation coefficients. In the result section, there is no any analysis and discussion about the Figures 2b and 2c. Please add more*

*illustration and discussion.]*

**Reply:** Many thanks for the referee's discussion. I an sorry about the description of *lag* correlation coefficients. Figure 2b and 2c show the spatial distributions of correlation coefficients of low cloud cover over the TP and the global specific humidity in the same month in summer (June, July and August separately) from 1979 to 2018 at (b) 400 hPa and (c) 500 hPa. We have rewritten the description and added the illustration in the manuscript (lines 127~139) as follow:

"The vertical section of the correlation coefficients along the south-north direction between the low cloud cover on the TP and the global water vapor are presented in Figure 1b. The obviously upward movement of water vapor over the TP can be seen in Figures 2a. It could be noticed that there exist the structures similar with the massive chimney between the convective cloud and the water vapor on the TP (Figures 1b and 2a). Figures 2b and 2c show significant correlation between convective clouds over the AWT and the changes of global water vapor from 1979 to 2018. Significant correlations extend from the TP southward and northward in the upper troposphere. It is remarkable that the high correlation areas passing the 90% confidence level expand towards the polar regions of both the southern and the northern hemispheres (Figures 1b, 2b and 2c), depicting the relation between the convective clouds and the global water vapor in the upper troposphere across the northern and southern hemispheres for an implication of the TP to global climate change. "

*[3. As seen from Figure 4, it contains lots of information, but the related analysis is too simply. Please add more analysis and discussions.]*

**Reply:** Many thanks for the referee's discussion.We added two subgraphs in Figure 4. The new

figure and description (lines 199-208) was adjusted as follow:

[Figure]

**Figure 4** The vertical sections of (a,d) vertical motion (contours, in unit: $10^{-2}$Pa·s$^{-1}$) and clolumn $Q_1$ (color contours, in unit:$10^{-3}$w kg$^{-1}$); (b,e) The vertical motion (contours, in unit: $10^{-2}$ Pa·s$^{-1}$) and correlation coefficients (color contours) between $Q_1$ and the vorticity as well as (c,f) vertical motion (contours, in unit: $10^{-2}$Pa·s$^{-1}$) and the correlation coefficients between $Q_1$ and the divergence (contours) in the TP,with Figs. a, b and c along 32 °N, and Figs. d, e and f along 95 °E. The green triangles indicate the AWT core region.

"Through the correlation analysis of the column apparent heat $Q_1$ over the TP as well as the three-dimensional structure of vorticity and divergence, it can be found that the apparent heat source $Q_1$ in the TP is an important forcing factor (Figure 4). The results show that the air heat island in the AWT is located at 300-500 hPa in the upper tropopshere, which is regarded as the high apparent heat $Q_1$ area significantly related to the convective clouds and the strong ascending movement (Figures 4a and 4d). Figures 4b, 4c, 4e and 4f present the correlations of the column apparent heat $Q_1$ in AWT with the divergence and vorticity fields over the TP, which can describe the effective "suction effect" with divergence (negative vorticity) at upper levels and convergence

(positive vorticity) at lower levels in the troposphere. "

*[4. L70, tropophere-> troposphere*

*5. L88, 100hpa--> 100 hPa*

*6. L307, 60oE - 180oE--> 60ºE - 180ºE*

*7. L142, the Asian water tower (AWT) --> AWT*

*8. L149, Figure 3c should be Figure 3b*

*9. L150, Figure 3d should be Figure 3c.]*

**Reply:** Following this comment, we have adjusted them as required.

*[10. P17, what does the shading mean in Figures 3b,c and d? What's the difference between Figure 3d and Figures 3b and c? The correlation in Figures 3b and c are based on the period of 1979-2016, aren't they? And why the correlation based on the period of 2014-2016 are given in particular?]*

**Reply:** Many thanks for the referee's discussion. The shaded parts in Figures 3b and 3c indicate correlation coefficients of TP-column Q1 integrated to water vapor.

The order of sub-graphs in Figure 3 has been changed. The new Figure 3b , 3d and 3e are adjusted as follow:

[Figure]

**Figure 3** (b) correlation vectors of the column $Q_1$ integrated over the TP region (80-102°E; 30-37.5°N) with the 300hPa vapor transport flux in July of 2014-2016,The shaded area indicates the correlation coefficient passing the the 90% confidence level; the correlation field between the total apparent heat source Q1 over the TP region (80-102°E; 30-37.5°N) with the water vapor (shaded) and water vapor flux (stream lines) in the surface layer (d) and middle layer (500hpa) (e) in summer over 1979-2015, respectively

The correlation mentioned in Figures. 3d and 3e is based on monthly mean Q1 and water vapor flux during 1979-2015 in summer. While Figure 3b shows correlation between daily mean Q1 in TP and water vapor flux in July from 2014 to 2016 at 300hPa so as to discuss the driving effect of Q1 on water vapor transport at the synoptic system and process scale in the Plateau region. We have supplemented the illustration in the manuscript as follow:

"The strong anticyclone in the upper troposphere over the southeastern TP takes a significant part in the upward transport of water vapor in the troposphere and stratosphere (Garny, et al., 2016; Fu, et al., 2006).  In order to understand the effect of the vertical transport window of troposphere over the TP on the global water vapor distribution from the perspective of the dynamic effect of anticyclone over the plateau driven by the heat sources, we presented the distributions of correlation coefficients between daily mean $Q_1$ in the TP and global water vapor flux in July from 2014 to 2016 at 300hPa (Figure 3b) Driven by the heat source of the TP, the anticyclone is formed in the upper troposphere over the TP and surrounding regions, which governed the water vapor transport form the TP not only to the surrounding area, but also extending to the north and south poles along the long-range transport channels (Figure 3b), which indicates the vertical transport window effect of the TP on global water vapor transport, especially over high-latitude regions such as the Arctic and Antarctic."

*[11. P18, the caption makes reader confused. All the contours in these four figures represent the vertical motion, not just in (a). Please rewrite the description of these four subgraphs.]*

**Reply:** Many thanks for the referee's suggestion. Following this comment, we have adjusted it as required.

"**Figure 4.** The vertical sections of (a,d) vertical motion (contours, in unit: $10^{-2}$Pa·s$^{-1}$) and clolumn $Q_1$ (color contours, in unit:$10^{-3}$w kg$^{-1}$); (b,e) The vertical motion (contours, in unit: $10^{-2}$ Pa·s$^{-1}$) and correlation coefficients (color contours) between $Q_1$ and the vorticity as well as (c,f) vertical motion (contours, in unit: $10^{-2}$Pa·s$^{-1}$) and the correlation coefficients between $Q_1$ and the divergence (contours) in the TP,with Figs. a, b and c   along 32 °N, and Figs. d, e and f along 95 °E. The green triangles indicate   the AWT core region."

**Reply to Referee 2**

**We are grateful to the referee for the encouraging comments and careful reviews which helped to improve the quality of our paper. In the followings we quoted each review question in the square brackets and presented our response after each paragraph.**

*[Review Comment: This paper investigates the effects of the Tibetan Plateau on the water vapor transport in the atmosphere and found that a summertime "hollow wet pool" and a vertical transport window exist in the troposphere over the Tibetan Plateau (TP) which have significant impacts on the global water vapor distribution. The results presented in this study are interesting and the content of the manuscript is well within the scope of ACP. However, the manuscript needs some revisions before it is accepted for publication in ACP. ]*

**Reply:** Thank you for the encouraging comments.

**Major comments:**

*[My first concern is about the causal relationship. Based on the correlation analysis, the authors argued that the effect of TP's vertical transport window of tropospheric vapor have impacts on global water vapor distribution, even the remote regions like the Arctic, Antarctic. However, correlation analysis alone can not reveal the causal relationship. I would suggest a model simulation with a passive tracer released over the TP to verify transport pathways of water vapor over the TP as suggested by the correlation analysis.]*

**Reply:**

Many thanks for the referee's suggestions. To verify transport pathways of water vapor over

the TP as suggested by the correlation analysis, we have used the methods of composite analysis to further understand the AWT heat source driving and maintaining water vapor transport from the TP to the high-latitude regions like the Arctic, Antarctic with the global influence. In the revised manuscript (lines 149-171) we have added the following discussions:

"The strong anticyclone in the upper troposphere over the southeastern TP takes a significant part in the upward transport of water vapor in the troposphere and stratosphere (Garny, et al., 2016; Fu, et al., 2006).    In order to understand the effect of the vertical transport window of troposphere over the TP on the global water vapor distribution from the perspective of the dynamic effect of anticyclone over the plateau driven by the heat sources, we presented the distributions of correlation coefficients between daily mean $Q_1$ in the TP and global water vapor flux in July from 2014 to 2016 at 300hPa (Figure 3b) Driven by the heat source of the TP, the anticyclone is formed in the upper troposphere over the TP and surrounding regions, which governed the water vapor transport form the TP not only to the surrounding area, but also extending to the north and south poles along the long-range transport channels (Figure 3b), which indicates the vertical transport window effect of the TP on global water vapor transport, especially over high-latitude regions such as the Arctic and Antarctic. To further verify the global transport pathways of water vapor from the TP, we used the methods of composite analysis to characterize global distribution of water vapor transport fluxes at the 300hpa in the years to anomalously high and low $Q_1$ over the TP. The TP's anticyclone in the upper troposphere is often associated with deep convection in the troposphere (Garny, et al., 2016). Figure 3c shows that in years with higher $Q_1$, stronger anticyclone formed at the upper troposphere (Figure 3b), which maintains the upward transport of water vapor to the upper troposphere, with strong transport of water vapor transport

the arctic and antarctic (Figure 3c), confirming the impact of the vertical transport in the troposphere driven by heat released within AWT in the TP on global water vapor transport especially to the polar regions."

[Figure]

**Figure 3** (b) correlation vectors of the column $Q_1$ integrated vertically over the TP region (80-102°E; 30-37.5°N) with the 300hPa vapor transport flux in July of 2014-2016, The shaded area indicates the correlation coefficient passing the 90% confidence level;(c) the difference of specific humidity (shading, unit:kg/kg) at 300 hPa in summer in 1998 and 2007 with anomalously high $Q_1$ and in 1997 and 2003 with anomalously low $Q_1$ in the AWT. The black and orange arrows

indicate respectively the anticyclonic circulations in the TP and water vapor transport pathways

from the TP to the Arctic and Antarctic regions.

*[another issue is the role of the TP's thermal effect on the formation of the transport channel of the*

*water vapor. It is proposed in the manuscript that the TP's thermal effect could make a strong*

*warm wet vapor transport channel connecting the water vapor source in the low latitude tropical*

*ocean. This conclusion is again drawn mostly from correlation analysis. Is it possible to do a few*

*sensitivity experiments with a numerical model to verify that the proposed transport channel is*

*indeed forced or maintained by the apparent heat source of the TP? Alternatively, it is better to*

*perform a composite analysis with respect to high and low Q to see whether this transport channel*

*will change with Q.]*

**Reply:**    Following the referee's suggestion, FLEXPART trajectory model is used to prove the

influence of the TP's Q1 on the water vapor transport channel connecting the TP to low latitude

ocean moisture source, and composite analysis is employed to further verify this with adding two

sub-graph as Figure 3f and Figure 3g and the content of Section 3.2 (Lines 181-191) and Section

3.3 (Lines 213-218) have adjusted with following sentences:

[Figure]

**Figure 3** (f) the backward trajectories of water vapor transport simulated with the model

FLEXPART in July, 2009. (g) the difference of vapor transport flux at 500 hPa (vectors, unit:$gs^{-1}hPa^{-1}cm^{-1}$) and specific humidity (color contours, unit:kg/kg) between summers with anomalously high $Q_1$ in 1998, 2005, 2007, 2008 and 2009 and with anomalously low $Q_1$ in 1994, 1997, 2001, 2002 and 2003 over the TP

"FLEXPART trajectory model (Stohl, et al., 2005;Reale,et al 2001; James, et al, 2004) was used to simulate the spatial and temporal changes of water vapor transport to the TRSR over the TP, driven with the ERA-Interim reanalysis data of meteorology with horizontal resolution of $0.75^o \times 0.75^o$ in July 2009. In the FLEXPART particle diffusion model, the 80000 particles was released at the TRSR (90°-102°E and 30°-35°N). In Figure 3f, it can be found that the water vapor in the TRSR was traced to water vapor source on the tropical Indian Ocean. The main water vapor from the central Indian Ocean in the southern hemisphere can be transported along the Somali jet flow through the Arabian Sea to the TP. The water vapor from the South China Sea and the Bay of Bengal was transported to the TP converging over the TRSR (Figure 3f), characterizing the water vapor transport channel from the southern hemispheric and low latitude oceans to the TP.

Figure 3g shows the difference of vapor transport flux and specific humidity at 500hPa in summer between anomalously high and low $Q_1$. When the $Q_1$ in TRSR is anomalously high, large water vapor from the tropical oceans is transported across the Bay of Bengal and the Indian peninsula, and entered the TP from the southern edge, revealing the TP's thermal effect could make a strong vapor transport channel connecting the water vapor source in the low latitude tropical oceans. "

We have accordingly cited the following article in the revised manuscript:

"Stohl, A., Forster, C., Frank, A., et al.: Technical note: The Lagrangian particle dispersion model

FLEXPART version 6.2. Atmos. Chem. Phys., 2005, 5, 2461–2474.

Reale, O., Feudale, L., Turato, B.: Evaporative moisture sources during a sequence of floods in the

Mediter-ranean region. Geophys Res Lett, 2001, 28, 2085–2088.

James, P., Stohl, A., Spichtinger, N.: Climatological aspects of the extreme European rainfall of

August 2002 and a trajectory method for estimating the associated evaporative source regions.

Nat Hazards Earth Syst Sci, 2004, 4, 733–746."

*minor comments:*

*[Title: 'global change' covers a relatively wide discipline. I would suggest change it to 'global*

*climate change'.]*

**Reply:** Following this comment, we have changed "global change" to "global climate change" in

the revised manuscript.

*[Line 41: 'The observed "CISK-like mechanism' may need a reference.]*

**Reply:** In the revised manuscript, we have added a reference as follows:

"The observed "CISK-like mechanism" is an important mechanism sustaining the atmospheric

"water tower" over the TP (Xu et al., 2014)

Xu, X, Zhao, T, Lu C., Guo, Y., Chen, B., Liu, R., Li, Y., and Shi, X.   (2014).An important mechanism

sustaining the atmospheric "water tower" over the Tibetan Plateau. *Atmos. Chem. Phys.*14:

11287-11295.https://doi.org/10.5194/acp-14-11287-2014"

*[Line 65: 'not enough attention' >> 'inadequate attention']*

**Reply:** We have changed to "inadequate attention" in the revised manuscript.

*[Line 71: What is the meaning of 'special column constructor'?]*

**Reply:** The 'special column constructor' means the vertical transport of water vapor in the troposphere constructed with the special column of apparent heat source in the AWT over the TP, which has been changed in the revised manuscript.

*[Page 5: some letters and symbols in the text which are used in the formulas should be italics.]*

**Reply:** Thanks for the careful review. They have been changed in the revised manuscript.

*[Line 103: 'productions' >>'products']*

**Reply:** We have corrected it in the revised manuscript.

*[Line 134-135: which variable can represent 'convective cloud activities?]*

**Reply:** we use the low cloud fraction to represent convective cloud activities based on the could characteristics observed in the TP, which has been added in the revised manuscript.

**Reply to the Community comment**

*[May I ask which tool/software can draw the diagram figure of Fig. 6 with all the clouds, water droplets and terrain]*

**Reply:** Thank you for your attention. Fig. 5 were drawn by NCL (NCAR Command Language v6.4.0) and the clouds, water droplets and terrain in Fig. 6 were drawn with Photoshop.